# Responding to preconditioned cues is devaluation sensitive and requires orbitofrontal cortex during cue-cue learning

Evan E Hart[1]*, Melissa J Sharpe[1,2], Matthew PH Gardner[1], Geoffrey Schoenbaum[1,3,4,5]*

[1]National Institute on Drug Abuse Intramural Research Program, National Institutes of Health, Baltimore, United States; [2]Department of Psychology, University of California, Los Angeles, Los Angeles, United States; [3]Department of Neuroscience, Johns Hopkins School of Medicine, Baltimore, United States; [4]Department of Psychiatry, University of Maryland School of Medicine, Baltimore, United States; [5]Department of Anatomy and Neurobiology, University of Maryland School of Medicine, Baltimore, United States

**\*For correspondence:**
evan.hart@nih.gov (EEH);
geoffrey.schoenbaum@nih.gov (GS)

**Abstract** The orbitofrontal cortex (OFC) is necessary for inferring value in tests of model-based reasoning, including in sensory preconditioning. This involvement could be accounted for by representation of value or by representation of broader associative structure. We recently reported neural correlates of such broader associative structure in OFC during the initial phase of sensory preconditioning (*Sadacca et al., 2018*). Here, we used optogenetic inhibition of OFC to test whether these correlates might be necessary for value inference during later probe testing. We found that inhibition of OFC during cue-cue learning abolished value inference during the probe test, inference subsequently shown in control rats to be sensitive to devaluation of the expected reward. These results demonstrate that OFC must be online during cue-cue learning, consistent with the argument that the correlates previously observed are not simply downstream readouts of sensory processing and instead contribute to building the associative model supporting later behavior.

## Introduction

Model-based reasoning requires using information about the structure of the world to infer information on-the-fly. This ability relies on a circuit involving the orbitofrontal cortex (OFC), as evidenced by the finding that humans and laboratory animals cannot infer value in several situations when OFC function is impaired (*Gallagher et al., 1999*; *Izquierdo et al., 2004*; *McDannald et al., 2005*; *Takahashi et al., 2009*; *West et al., 2011*; *Gremel and Costa, 2013*; *Reber et al., 2017*; *Howard et al., 2020*). Model-based reasoning can be isolated in sensory preconditioning (*Brogden, 1939*). During this task, subjects first learn an association between neutral or valueless stimuli. If one of these stimuli is thereafter paired with a biologically meaningful stimulus (e.g. food), presentation of the never-reward-paired preconditioned stimulus will produce a strong food-approach response. This demonstrates that subjects formed an associative model in which the preconditioned cue predicts the conditioned cue, which can subsequently be mobilized to produce a food-approach response when the preconditioned cue is presented. Because the preconditioned cue itself was never reinforced, this response to the preconditioned cue cannot be explained by model-free

mechanisms. Preconditioned responding is reflective of the ability of subjects to infer value on-the-fly using knowledge about the associative structure of the information acquired during preconditioning.

Previous work from our lab has shown that the lateral OFC activity is necessary for inferring the value of never-reinforced sensory stimuli in the probe test at the end of sensory preconditioning (*Jones et al., 2012*). Interestingly, subsequent single-unit recording showed that neural activity in the OFC reflects not only the value predictions in the final probe test, but also tracks acquisition of the sensory-sensory associations in the initial phase of training (*Sadacca et al., 2018*). This result suggested a wider role for OFC in building associative models, independent of encoding of overt biological significance or value, the role often ascribed to this region (*Kringelbach, 2005*; *Padoa-Schioppa and Assad, 2006*; *Padoa-Schioppa, 2011*; *Levy and Glimcher, 2012*).

Of course, single-unit recording is correlative and cannot demonstrate that OFC activity is necessary for proper acquisition of the sensory-sensory associations during preconditioning. The correlates could simply reflect activity in primary sensory (*Engineer et al., 2008*; *Garrido et al., 2009*; *Headley and Weinberger, 2015*) or other cortical areas that are necessary in the first phase (*Port et al., 1987*; *Robinson et al., 2011*; *Robinson et al., 2014*; *Fournier et al., 2020*). If this is the case, then interference with processing in OFC during preconditioning should not affect responding to the preconditioned cue during the probe test. On the other hand, if the representations of the sensory-sensory associations formed in OFC in the initial phase contribute to building the model of the task used to infer value and drive responding in the probe test, then interference with processing in OFC during preconditioning should affect probe test responding, either directly impairing normal responding to the preconditioned cue, if responding in our task is entirely model-based as we have suggested (*Jones et al., 2012*; *Sadacca et al., 2018*), or by making any responding insensitive to devaluation of the predicted food reward, if responding in the probe is partly maintained by transfer of cached value to the cue during conditioning (*Doll and Daw, 2016*). Here we tested for such functional effects by training rats on a sensory preconditioning task nearly identical to those previously used by our lab (*Jones et al., 2012*; *Sharpe et al., 2017*; *Sadacca et al., 2018*), optogenetically inactivating the lateral OFC during cue presentation in the preconditioning phase of training and adding an additional assessment of responding after devaluation at the end of the experiment. We found that interfering with processing in OFC during the preconditioning phase completely abolished responding to the preconditioned cue in the probe test after conditioning, responding that in controls was subsequently disrupted by devaluation of the expected food reward.

## Results

### Orbitofrontal cortex activity is necessary for forming sensory associations

We tested whether the OFC has a causal role in learning about value-neutral associations acquired during sensory preconditioning. For this, we used a sensory preconditioning task largely identical to that used in previous work (*Figure 1A*) and optogenetic inhibition (*Figure 1A–C*) to test whether this activity during preconditioning is necessary for proper value inference during later probe testing.

Prior to any training, two groups of rats underwent surgery in which virus was infused into the lateral area of OFC, where recordings were obtained in our prior experiment (*Sadacca et al., 2018*). One group, the NpHR experimental group (N = 21), received AAV5 containing halorhodopsin (NpHR) linked to a CamKIIα promoter. NpHR is a light gated chloride pump that induces neuronal hyperpolarization to allow real-time neuronal inactivation (*Gradinaru et al., 2008*). The other group of rats, the eYFP control group (N = 22), received the same virus but without the opsin. Both viruses contained eYFP for histological verification of successful transfection. All rats also had fiberoptic probes implanted bilaterally overlying the injection site to allow light delivery. After recovery from surgery and a brief period of food deprivation, training began.

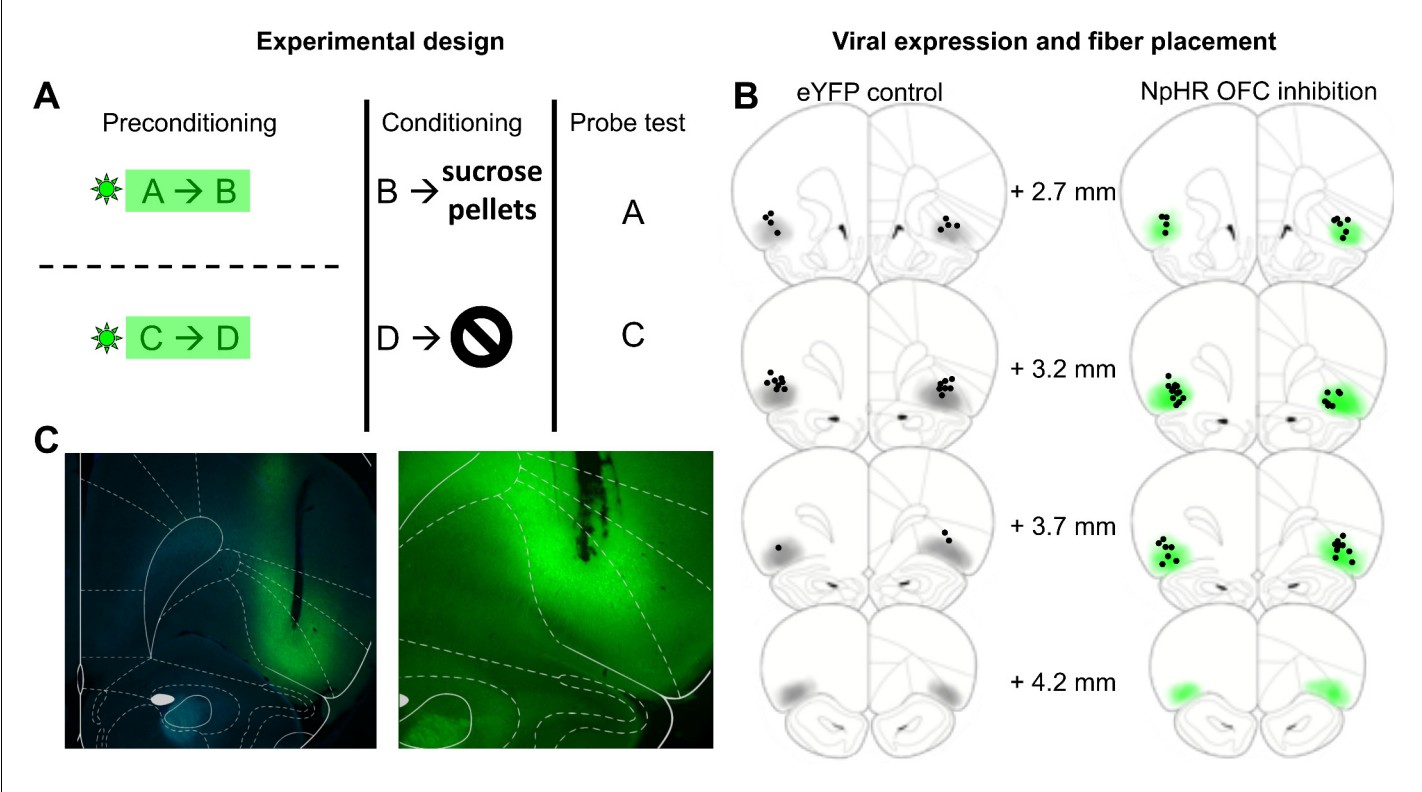

**Figure 1.** Task schematic and histology. (A) Task Schematic. Rats received value-neutral cue pairings during preconditioning. Light was delivered to eYFP control and NpHR rats during cue presentation. Reward conditioning occurred in phase two during which cue B was paired with sucrose pellets and Cue D was presented alone. Cues A and C were probed in phase 3. (B) Schematic reconstruction of fiber tip placements and viral expression in eYFP controls (left) and NpHR rats (right). Light shading indicates areas of minimal spread and dark shading areas of maximal spread. Black circles indicate fiber tip placements. (C) Representative photomicrographs of fiber placements and eYFP expression in lateral OFC.

## Preconditioning

Training began with a preconditioning phase, which involved two sessions in which the rats were presented with neutral cue pairs (A→B; C→D; six pairings of each per session), in a blocked and counterbalanced fashion. Laser light was delivered into OFC in both eYFP controls and NpHR rats, starting 500 ms prior to presentation of the first cue in each pair and lasting for the duration of both cues. No reward was presented, so responding to the food cup during this phase was negligible, and there were no differences between cues or groups (*Figure 2A*). A two-way ANOVA with cue as a within subjects factor and group as a between subjects factor revealed no main effects of cue ($F_{(3,123)}=0.43$, p=0.73) or group ($F_{(1,41)}=1.99$, p=0.17) and no group x cue interaction ($F_{(3,123)}=0.86$, p=0.46).

## Conditioning

Following preconditioning, rats underwent six sessions of Pavlovian conditioning in which they received cue B paired with reward and cue D without reward (B→US; D; six pairings of each per session, interleaved in a counterbalanced fashion). As expected, rats developed a conditioned response to B with training, responding more at the food cup during cue B than during cue D, and there was no difference between groups (*Figure 2B*). A three-way mixed-design (cue, session within subjects factors; group between subjects factor) ANOVA revealed significant main effects of cue ($F_{(1,41)}=143.00$, p<0.0001) and session ($F_{(5,205)}=36.23$, p<0.0001), and a significant session x cue interaction ($F_{(5,205)}=9.76$, p<0.0001). Critically, there were no main effects of, nor any interactions with, group (main: $F_{(1,41)}=0.02$, p=0.90; session x group: $F_{(5,205)}=1.72$, p=0.13; group x cue interaction: $F_{(1,41)}=0.53$, p=0.47).

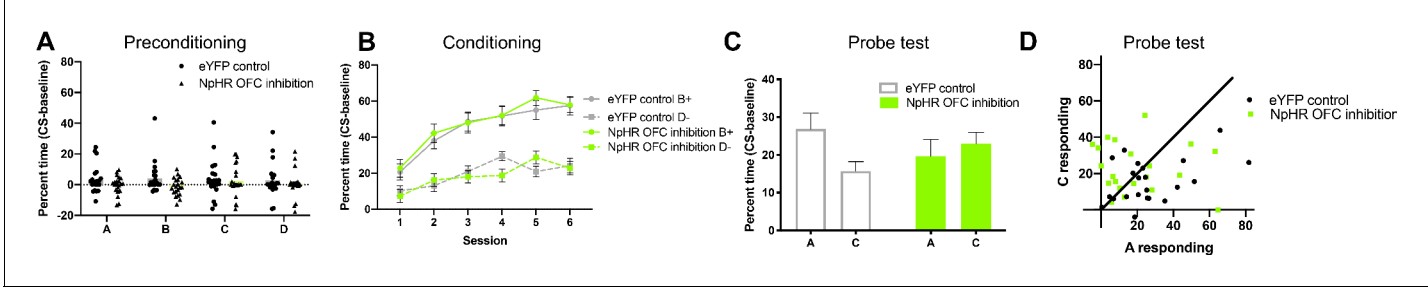

**Figure 2.** OFC inhibition during preconditioning impairs value inference during the probe test. (A) All rats showed low levels of responding during preconditioning when neutral cue pairs were presented. There was no effect of cue or group. (B) During reward conditioning, all rats (grey, eYFP; green, NpHR) showed higher responding to the rewarded cue B (solid lines) than to the non-rewarded cue D (dotted lines). (C) During the probe test, responding to A was higher than to C in control rats (eYFP, left, grey), but not in OFC inhibition rats (NpHR, right, green). (D) Scatter plot of A versus C responding during the probe test. To the extent that responding to A and C are equal, points should congregate along the diagonal. Points below the diagonal indicate A>C responding and therefore evidence of preconditioning.

The online version of this article includes the following source data for figure 2:

**Source data 1.** Source data for *Figure 2*.

### Probe test

Finally, the rats underwent a single probe test in which cues A and C were presented alone and without reward, counterbalanced across subjects. We found that control rats responded more to A than C, and NpHR rats did not (*Figure 2C–D*). A two-way mixed-design ANOVA revealed no main effects of cue ($F_{(1,41)}$=1.33, p=0.26) or group ($F_{(1,41)}$=0.00, p=0.99). There was a significant cue x group interaction ($F_{(1,41)}$=4.57, p=0.04); ergo, the effect of cue during the probe test (A-C) did depend on group (eYFP control – NpHR OFC inhibition). Posthoc comparisons revealed responding to A was significantly higher than C in controls (p=0.046), but not in NpHR rats (p=0.75).

## Preconditioned responding in controls is sensitive to outcome devaluation

### Taste aversion training

To confirm the associative basis of responding to the preconditioned cue observed in controls, a subset of the eYFP rats (n = 13) were split into two groups (paired and unpaired) with equal

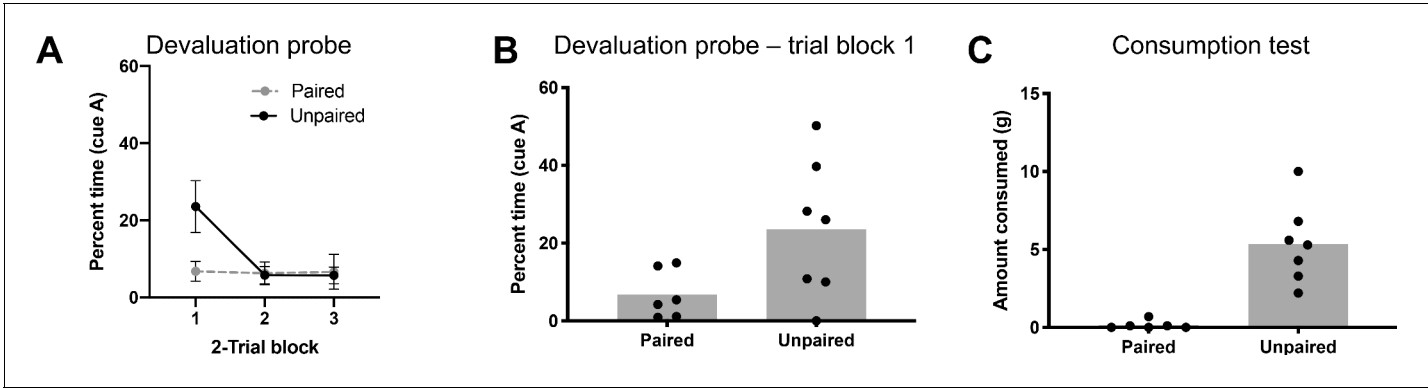

**Figure 3.** Responding to preconditioned cues in control rats is outcome devaluation sensitive. (A) Rats that received LiCl and pellets unpaired (black) showed higher levels of responding to A during the first 2-trial block than rats that received them paired (grey). All rats quickly extinguished. (B) Average responses during the first 2-trial block. (C) Rats that received paired LiCl and pellets (left) consumed fewer pellets than rats that received them unpaired (right) during the consumption test.

The online version of this article includes the following source data for figure 3:

**Source data 1.** Source data for *Figure 3*.

responding to A and C (no main effect of group ($F_{(1,11)}$=0.06, p=0.81) or group x cue interaction ($F_{(1,11)}$=2.60, p=0.13)). All rats consumed all pellets during the first free exposure session. Following this, the paired group immediately received lithium chloride injections, while the unpaired group received the same injections 24 hr following sucrose pellet exposure. Subsequently, during the second free exposure session, the paired group demonstrated a conditioned taste aversion, consuming significantly fewer pellets than in the first test ($t_{(5)}$ = 4.95, p=0.001, second exposure mean = 2.25 g, range = 0.6–3.0 g); to equalize exposure, rats in the unpaired group were given sucrose pellets yoked to the amount that the paired group consumed in this second session, then rats in both groups underwent probe testing.

## Devaluation probe

For the devaluation probe test, rats in both the paired and unpaired groups were presented with cue A six times, alone and without reward, after which they were given a final consumption test to confirm devaluation. On probe, responding extinguished quickly in both groups, however it was much higher in the unpaired group on the initial trials (*Figure 3A*). A two-way ANOVA revealed a significant main effect of trial block ($F_{(2,22)}$=4.91, p=0.02) and significant trial block x group interaction ($F_{(2,22)}$=4.58, p=0.02). Post hoc comparisons revealed responding was significantly higher to A in the unpaired group than the paired group during the first trial block (p=0.02; *Figure 3B*). This change in responding was accompanied by reduced consumption of the outcome by rats in the paired group ($t_{(11)}$ = 4.95, p=0.0004; *Figure 3C*).

## Discussion

While the OFC is generally thought to contribute to behavior where there are changes in, or computations of, value (*Gallagher et al., 1999*; *Kringelbach, 2005*; *Padoa-Schioppa and Assad, 2006*; *Schoenbaum et al., 2011*), our lab recently showed that neural activity in lateral OFC can also track the development of valueless sensory-sensory associative information in a sensory preconditioning task (*Sadacca et al., 2018*). While performance in the final phase of this task is ultimately sensitive to OFC manipulations in both rats (*Jones et al., 2012*) and humans (*Wang et al., 2020b*), the development of neural correlates of neutral sensory associations in the initial phase of training suggested that OFC might not be restricted to encoding information once clear biological significance is established and instead could be more widely involved in representing structure among features of the environment, with value being just one of these features. Here we tested this question by optogenetically-inactivating the OFC during the initial phase of preconditioning, when correlates were previously observed, and we found that this manipulation also disrupted performance in the final phase of the task. These findings are consistent with the argument that the associative correlates we observed during preconditioning in our prior study are not simply downstream reflections of sensory or other cortical processing and instead must contribute to the construction of the model used to guide responding in the final probe test. These data are also in accord with the findings that OFC represents not only value, but also features of the environment such as trial type (*Lopatina et al., 2017*), sequence (*Zhou et al., 2019*), direction, and goal location (*Feierstein et al., 2006*; *Mainen and Kepecs, 2009*).

These results come with several important considerations that are worth discussing. The first is whether OFC inhibition caused the rats to simply generalize between the preconditioned cues. In considering this possibility, we would first note that generalization or failure to treat these cues differently would be to some extent the final common outcome of disrupting the more detailed encoding of their identities and different associative relationships to reward that we argue is affected. For that reason, it is difficult to formally rule this out. However we believe that generalization as a primary cause is unlikely given the low overall levels of responding to these cues during the probe, relative to responses to the reward-paired cue. Additionally, responding to the two preconditioned cues in the experimental group was no different than responding to the preconditioned cue predicting reward in the control group. If generalization alone accounted for the results, we would expect higher levels of responding to both cues during the probe in the OFC-inhibited rats. There is also no evidence from other studies that the OFC is necessary for discriminating the sensory properties of cues; rats without OFC or in whom the OFC is inactivated perform well in a number of settings that

require them to distinguish different cues to guide responding (*Schoenbaum et al., 2002*; *Boulougouris et al., 2007*; *Takahashi et al., 2009*; *Jones et al., 2012*).

A second consideration to discuss is the timing of our manipulation and its specificity. We inhibited OFC only during cue presentation, precisely when we previously observed single unit correlates of cue-cue encoding; our design did not include control conditions in which similar inhibition was done before or after the cues or during the intertrial intervals. Thus, while we can conclude that OFC must be online at the time the neural correlates were observed, we cannot conclude that it does not need to be online at any other time during the first phase. Although OFC can be inactivated briefly between trials without affecting OFC-dependent learning and behavior (*Takahashi et al., 2013*), we would not be surprised if processing outside the strictly defined cue period – say immediately before or after – were also necessary for normal learning. Such a result would not invalidate what we have shown here, but it would show that learning is not fully completed during the external sensory events. While defining this critical period would be very interesting, it would require a succession of control groups that we did not deem practical (just after the cues, 20 s after the cues, 40 s after the cues, and so on).

A third consideration is whether the effect of optogenetic inhibition could reflect something other than the inhibition of the neural correlates demonstrated in our earlier study. While we know from our control group, that simply delivering light during the cue period is not sufficient, perhaps inhibiting OFC neurons during this period distracts the rats, because it is rewarding or punishing or for some other reason. If this were the case, then we would have expected to see an impact of this on learning and responding to these cues in the later periods of the task. We did not see any evidence of this. Further we know from a number of prior studies that brief periods of OFC inhibition, similar to that employed here, often have minimal or no effects on ongoing behavior (*Takahashi et al., 2013*; *Gardner et al., 2017*; *Gardner et al., 2018*). These results suggest this is not, by itself, rewarding, painful or otherwise distracting. Finally, it has been shown that lateral OFC inactivation is not sufficient to produce state-dependent learning, which provides very strong evidence that it does not create a tangible experience in and of itself (*Panayi and Killcross, 2014*).

A final consideration is that all rats received food cup approach shaping prior to preconditioning. It is possible that this changes the salience of the conditioning chamber context and causes increased attention to the cues during preconditioning, which may facilitate learning. If this is the case, then one interpretation of the current findings would be that OFC is necessary for normal acquisition of the associations during preconditioning because it contributes to this boost in attention, perhaps secondary to a role in processing the value of the context, rather than contributing to the encoding itself. This idea would be consistent with results showing correlates of risk (*O'Neill and Schultz, 2010*; *Ogawa et al., 2013*), uncertainty (*van Duuren et al., 2009*) and salience (*Ogawa et al., 2013*) in OFC neurons, as well as the finding that OFC value coding is attentionally gated (*McGinty et al., 2016*) and active in auditory oddball task (*Nguyen and Lin, 2014*). However, while it is possible that inactivating OFC affected learning of the sensory-sensory associations by interfering with a primary role for OFC in supporting the context or even cue salience, it seems more parsimonious to us to propose that the role OFC plays in supporting salience is secondary to the associative learning role that OFC clearly supports. This idea is consistent with attentional models, both old and new, that link cue salience to associative strength – changes in cue associative strength drive changes in salience (*Mackintosh, 1975*; *Pearce and Hall, 1980*; *Haselgrove et al., 2010*; *Esber and Haselgrove, 2011*).

Of course, we do not intend to suggest that OFC is the only area that is necessary for this sort of value-neutral sensory learning. This is surely not the case. Indeed even when preconditioning is the readout, we know that interfering with several cortical areas including hippocampus (*Port et al., 1987*), retrosplenial cortex (*Robinson et al., 2011*; *Robinson et al., 2014*; *Fournier et al., 2020*), and perirhinal cortex (*Holmes et al., 2013*; *Wong et al., 2019*) seems to impair the acquisition of sensory-sensory associations. While most cortical and subcortical areas encode variables relevant to predicting biologically-relevant outcomes (*Milad and Quirk, 2002*; *Burgos-Robles et al., 2009*; *Kennerley et al., 2011*; *Cai and Padoa-Schioppa, 2012*; *Burgos-Robles et al., 2013*; *Giustino et al., 2016*; *Giustino et al., 2019*; *Halladay et al., 2020*), few studies look at whether the same areas track incidental sensory-sensory associations. Perhaps correlates like those observed in the OFC are ubiquitous; since the organism does not yet understand whether this information is useful it is encoded and is available for some time to enter into relationships reflecting the function of

the area. This account would predict that other cortical areas (e.g. prelimbic, infralimbic, cingulate, retrosplenial) will likely also exhibit correlates of incidental sensory-sensory pairings, like OFC, yet unique single unit responses would not be apparent until the probe test, when subjects must infer specific information about the preconditioned cues based on new information acquired during conditioning. Indeed, fMRI correlates of cue-cue learning were observed in precentral gyrus, middle occipital gyrus, insula, and hippocampus in a human sensory preconditioning task (*Wang et al., 2020a*), and single unit correlates also exist in nucleus accumbens (*Cerri et al., 2014*). Along similar lines, OFC might contribute something to how the sensory-sensory associations are laid down that is necessary for their subsequent recall for use in the probe test, when it involves appetitive responding, while not being involved more generally in all forms of sensory associations. This could be assessed with fear conditioning.

Lastly we would note that while preconditioned responding has been shown as sensitive to extinction of the conditioned cue in fear conditioning (*Rizley and Rescorla, 1972*) as well as to aversive conditioning when the conditioned cue is a tastant (*Blundell et al., 2003*), sensitivity of the preconditioned response to devaluation of the reward predicted by the conditioned cue is novel. This result is important because, while it is conciliant with a mechanism whereby preconditioned responding reflects inference about food in the probe test, it is less consistent with other mechanisms whereby the preconditioned cue has been proposed to acquire value directly through so-called mediated learning in the conditioning phase. Such directly acquired value should leave residual responding that is resistant to devaluation. We did not see any evidence of this, consistent with prior work showing that preconditioned cues trained as we have done here do not acquire value (*Sharpe et al., 2017*). The demonstration of the devaluation sensitivity of this response nicely supports our contention that its OFC-dependence, both here and in prior work, reflects the role of OFC in using the underlying associative model to infer the likely delivery of food.

## Materials and methods

### Subjects
Fifty-four adult male (n = 31) and female (n = 23) Long-Evans rats age three to four months weighing between 270–320 g during the time of experiments were individually housed and given ad libitum food and water except during behavioral training and testing. The effect of cue during the critical probe test did not depend on sex (main: $F_{(1,39)}$=0.54, p=0.47; sex x cue: $F_{(1,39)}$=0.19, p=0.66; sex x cue x group interaction: $F_{(1,39)}$=2.66, p=0.11), so male and female rats were pooled for all analyses presented in the results. One day prior to behavioral training rats were food restricted to 12 g (males) or 10 g (females) of standard rat chow per day, following each session. Rats were maintained on a 12 hr/12 hr light/dark cycle and tested during the light phase between 8:00 am and 12:00 pm five days per week. Experiments were performed at the National Institute on Drug Abuse Intramural Research Program, in accordance with NIH guidelines. Eleven subjects were not included in the final analyses due to loss of a headcap (n = 1), target miss and/or lack of viral expression (n = 7), and failure to acquire greater rewarded cue than non-rewarded cue responding during the conditioning phase (n = 3).

### Apparatus
All chambers and cue devices were commercially available equipment (Coulbourn Instruments, Allentown, PA). A recessed food port was placed in the center of the right wall of each chamber and was attached to a pellet dispenser that dispensed three 45 mg sucrose pellets (Bioserv F0023) during each rewarded cue presentation. Auditory cues consisted of a pure tone, siren, 2 Hz clicker, and white noise, each calibrated to 70 dB. Cues A and C were white noise or clicker; cues B and D were tone or siren; all cues were counterbalanced across rats, sexes, and groups such that every possible permutation of cue identities, sex, and group was equally present.

### Surgical procedures
Rats were anesthetized with isoflurane (3% induction, 1–2% maintenance in 2 L/m $O_2$) and placed in a standard stereotaxic device. Burr holes were drilled bilaterally on the skull for insertion of fiber optics and 33-gauge injectors (PlasticsOne, Roanoke, VA). Rats were infused with 1.0 µL of virus at a

flow rate of 0.2 μL/minute, and injectors were subsequently left in place for five additional minutes to allow for diffusion of solution. Halorhodopsin (NpHR), a light gated chloride pump that induces neuronal hyperpolarization (*Gradinaru et al., 2008*), was used for OFC inhibition. Enhanced yellow fluorescent protein (eYFP) was used as the control virus. eYFP rats received AAV5-CaMKIIα-eYFP (titer ~ $10^{12}$). NpHR rats received AAV5-CaMKIIα-eNpHR-eYFP (titer ~ $10^{12}$) (Gene Therapy Center, University of North Carolina Chapel Hill). The coordinates used for targeting OFC in male rats were: AP = +3.0 mm, ML=±3.2 mm, DV = −5.3 mm from Bregma. Female rats received infusions at: AP = +2.85 mm, ML=±3.04 mm, DV = −5.09 mm from Bregma. Optic fibers (200 μm core diameter, Thorlabs, Newton, NJ) were implanted bilaterally at 10 degrees away from the midline at the following target: AP = +3.0 mm, ML=±3.2 mm, DV = −4.90 mm from Bregma (males) or AP = +2.85 mm, ML=±3.04 mm, DV = −4.66 mm from Bregma (females). Headcaps were secured with 0–80 1/8' machine screws and dental acrylic. Rats received 5 mg/kg carprofen s.c. on the day of surgery and 60 mg/kg p.o. cephalexin for ten days following surgery to prevent infection. Rats were given three weeks to allow viral expression prior to testing.

## Behavioral training

The task used was nearly identical to those from previous studies (*Jones et al., 2012*; *Sharpe et al., 2017*; *Sadacca et al., 2018*).

### Preconditioning

On the day following the beginning of food restriction, rats were shaped to retrieve pellets from the food port in one session. During this session, twenty pellets were delivered, with each single pellet delivery occurring on a 3–6 m variable time schedule. After this shaping, rats underwent two days of preconditioning. In each day of preconditioning, rats received trials in which two pairs of auditory cues (A→B and C→D) were presented in blocks of six trials, and the order was counterbalanced across rats. Cues were each 10 s long, the inter-trial intervals varied from 3 to 6 m, and the order of the blocks was alternated across the two days. 532 nm laser light was calibrated to between 16 and 18 mW at patch cord output. Constant laser light began 500 ms prior to presentation of each cue pair and ended with cue termination. Cues A and C were a white noise or a clicker, counterbalanced. Cues B and D were a siren or a constant tone, counterbalanced. Behavioral responses reported are the percentage of time spent during each 10 s cue in the food port, minus the 10 s preceding each cue.

### Conditioning

Rats underwent conditioning following preconditioning. No laser light occurred during this phase. Each day, rats received a single training session, consisting of six trials of cue B paired with pellet delivery and six trials of D paired with no reward. The pellets were presented three times during cue B at 3, 6.5, and 9 s into the 10 s presentation. Cue D was presented for 10 s without reward. The two cues were presented in 3-trial blocks, counterbalanced across subjects, and with the order of blocks alternating across days. The inter-trial intervals varied between 3 and 6 min. Behavioral responses reported are the percentage of time spent during each 10 s cue in the food port, minus the 10 s preceding each cue.

### Probe test

After conditioning, the rats underwent probe testing. Rats were given blocked or alternating presentation of cues A and C alone, six times each, without reward. The order of the cues/blocks was counterbalanced across subjects. Inter-trial intervals were variable 3–6 m. Behavioral responses reported are the percentage of time spent during each 10 s cue in the food port, minus the 10 s preceding each cue.

### Taste aversion

Procedures followed those of previous studies (*Pickens et al., 2003*; *Singh et al., 2011*). Rats in the eYFP group were matched for performance and divided into paired and unpaired groups. Taste aversion training lasted four days during which both groups received two ten-m sessions of access to sucrose pellets and two injections of 0.3 M LiCl, 5 mL/kg, i.p. On days 1 and 3, rats in the paired

group were given 10 m free access to 4.5 g pellets in the testing chambers, immediately followed by injections. Rats in the unpaired group were given injections with no pellet exposure. On Days 2 and 4, rats in the unpaired group were given 10 m free pellet exposure in the chambers. All rats therefore received equal free exposure to pellets, the chambers, and injections; the groups only differed in contiguity between pellets and injections. Behavioral responses reported are the percentage of time spent during each 10 s cue presentation in the food port.

### Devaluation probe

Following taste aversion training, rats were given a devaluation probe test during which cue A was presented six times. The intertrial interval varied between 3 and 6 m. Immediately following this, rats were presented with 10 g free pellets in the testing chambers for 10 m. Behavioral responses reported are the percentage of time spent during each 10 s cue presentation in the food port.

### Histology

Rats were sacrificed by carbon dioxide and perfused with PBS followed by 4% formaldehyde in PBS. 0.05 mm coronal slices were visualized for confirmation of fiber tip placements and eYFP expression by a blind experimenter.

## Statistical analyses

Data were collected using Graphic State three software (Coulbourn Instruments, Allentown, PA). Raw data were processed in Matlab 2018b (Mathworks, Natick, MA) to extract percentage of time spent in the food port during and preceding cue presentation. These behavioral data were analyzed in Matlab and Graphpad Prism (La Jolla, CA). Responses during preconditioning and probe testing were compared by two-way (group, cue) ANOVA. Responses during conditioning were compared by three-way (group, cue, session) ANOVA. Posthoc comparisons using Šidák's correction for family-wise error (*Sidak, 1967*) were conducted on probe test and devaluation probe test data. Adjusted p values are reported.

## Additional information

### Competing interests

Geoffrey Schoenbaum: Reviewing editor, *eLife*. The other authors declare that no competing interests exist.

### Funding

| Funder | Grant reference number | Author |
| --- | --- | --- |
| National Institute on Drug Abuse | zia-da000587 | Geoffrey Schoenbaum |

The funders had no role in study design, data collection and interpretation, or the decision to submit the work for publication.

### Author contributions

Evan E Hart, Conceptualization, Formal analysis, Investigation, Methodology, Writing - original draft, Writing - review and editing; Melissa J Sharpe, Conceptualization, Investigation, Methodology, Writing - review and editing; Matthew PH Gardner, Conceptualization, Methodology, Writing - review and editing; Geoffrey Schoenbaum, Conceptualization, Supervision, Funding acquisition, Writing - original draft, Writing - review and editing

### Author ORCIDs

Evan E Hart (iD) https://orcid.org/0000-0003-1487-5628
Melissa J Sharpe (iD) http://orcid.org/0000-0002-5375-2076

Matthew PH Gardner (iD) http://orcid.org/0000-0002-9146-5043
Geoffrey Schoenbaum (iD) https://orcid.org/0000-0001-8180-0701

### Ethics

Animal experimentation: This study was performed in strict accordance with the recommendations in the Guide for the Care and Use of Laboratory Animals of the National Institutes of Health. All of the animals were handled according to approved institutional animal care and use committee (IACUC) protocols (#18-CNRB-108) of the NIDA-IRP. The protocol was approved by the ACUC at the IRP (Permit Number: A4149-01). Every effort was made to minimize suffering.

### Decision letter and Author response

Decision letter https://doi.org/10.7554/eLife.59998.sa1
Author response https://doi.org/10.7554/eLife.59998.sa2

## Additional files

### Supplementary files

• Transparent reporting form

### Data availability

All data generated are contained within the source data files for Figure 2 and 3.

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
