## [Decision Letter]

Thank you for submitting your article "Devaluation-sensitive responding to preconditioned cues requires orbitofrontal cortex during initial cue-cue learning" for consideration by *eLife*. Your article has been reviewed by Michael Frank as the Senior Editor, a Reviewing Editor, and two reviewers. The reviewers have opted to remain anonymous.

The reviewers have discussed the reviews with one another and the Reviewing Editor has drafted this decision to help you prepare a revised submission.

Summary:

Sensory preconditioning has been a powerful behavioral tool for exploring how, when and where model based associative structures are stored in the brain. In the present experiment, a natural follow-up to a recent study by Sadacca et al., (2018), the authors explore a single outstanding issue relating to whether neural encoding of valueless associations at the time of preconditioning are actually necessary for subsequent associative value transfer to non-conditioned cues. The broader issue here is whether OFC is a specialized economic structure or whether it plays a more general role in linking information about outcomes to options. Here they use the temporal specificity of optogenetics, activating halorhodopsin to inhibit the activity of OFC neurons only when the two unconditioned cues are presented. They report that inhibition during this period abolishes subsequent responding to the unconditioned cue, indicating that OFC inhibition blocked cue-cue learning. A subsequent experiment using only the control animals, showed that devaluing the US was associated with reductions in responding confirming that model-based associations formed during sensory preconditioning are dependent on OFC.

Essential revisions:

This is an elegant albeit short report that will be of interest to many in the field but mostly as an important control to a larger body of work. There are, however, a couple of places where the manuscript could use some work, most notably in clarifying what the authors think the mechanisms of sensory preconditioning are and confounds relating to the design.

1) The authors only delivered laser light to inhibit OFC neurons during the 10 seconds where the two cues were presented together. This makes perfect sense as they were aiming to understand how disrupting OFC at this point impacts learning. A question then logically follows; is it the case that disrupting OFC for this length of time, irrespective of when during the preconditioning phase is sufficient to disrupt associative learning? Put simply, if 10 seconds of laser light was delivered during the ITI in the preconditioning phase, is this sufficient to disrupt learning/retention of the cue-cue associations? I'm not asking for an extra experiment here, but this should be acknowledged in the discussion as it speaks to the role of OFC in both learning AND/OR retention of cue-cue associations.

2) The discussion of attention in the learning of cue-cue value associations should be expanded as this is the key take away from the paper. Cues that the rats did not expect to be presented are likely learned about as they are attentionally salient. More of a discussion of the role of OFC in attentionally salient events and interaction with other areas feels more than a little appropriate here. For instance, OFC is known to be modulated during odd-ball tasks through input from the basal forebrain (Nguyen and Lin, 2014) and more recently work in primates has started to reveal that OFC encoding of value is attentionally gated (McGinty, et al., 2015). Similarly, how does this speak to learning models based on attention such as the Pearce-Hall model?

3) The paper is brief and relies on a small amount of data. From what I can tell, the key analysis is Figure 2C, which is described in subsection “Orbitofrontal cortex activity is necessary for forming sensory associations”. The thing I don't totally get is that the finding relies on a positive effect in the control condition and a null effect in the experimental condition (subsection “Orbitofrontal cortex activity is necessary for forming sensory associations”). Strictly speaking, this is invalid – the difference between significant and non-significant is not in itself significant (it can be and often is, but isn’t necessarily so, Nieuwenhuis, Forstmann and Wagenmakers, 2011). This is especially important to address because the control effect is marginally significant (p=0.046). Anyway, it looks like it probably is significant, but that ought to be reported. (Another reviewer during consultation surmised that you did do this analysis but they weren't quite sure "It looks like they have computed the interaction term and reported it 'however, there was a significant cue x group interaction (F(1,41)=4.57, p=0.04).'".

4) The second issue is that a disruption of effect, while it does demonstrate causality, has multiple possible interpretations. It could be that the optogenetic effect is, like, super-distracting, or causes a headache or something, and that causes the animal to get distracted and not do very well on the task. There are probably ways to deal with this concern, like showing other things that aren't disrupted. But in this case, I want to see that because we don't want to assume causality from any abolition of effect.

---

## [Author Response]

Essential revisions:This is an elegant albeit short report that will be of interest to many in the field but mostly as an important control to a larger body of work. There are, however, a couple of places where the manuscript could use some work, most notably in clarifying what the authors think the mechanisms of sensory preconditioning are and confounds relating to the design.

We thank the editors and reviewers for their constructive feedback, which has greatly improved the manuscript. In response to this feedback, we have added considerable discussion of caveats, experimental design, elaboration on statistics, and expanded discussion of psychological processes that could mediate sensory preconditioning. We have also made minor changes to improve clarity – namely a change in the Title and added description of the experimental and control groups. In addition to this we have softened some of the language to reflect that it is now more established that OFC is not only engaged in value-guided learning and behavior, as a reviewer noted. Below we give our point-by-point responses.

1) The authors only delivered laser light to inhibit OFC neurons during the 10 seconds where the two cues were presented together. This makes perfect sense as they were aiming to understand how disrupting OFC at this point impacts learning. A question then logically follows; is it the case that disrupting OFC for this length of time, irrespective of when during the preconditioning phase is sufficient to disrupt associative learning? Put simply, if 10 seconds of laser light was delivered during the ITI in the preconditioning phase, is this sufficient to disrupt learning/retention of the cue-cue associations? I'm not asking for an extra experiment here, but this should be acknowledged in the discussion as it speaks to the role of OFC in both learning AND/OR retention of cue-cue associations.

We appreciate this concern and have added more discussion of this and other caveats to the Discussion section. Although we realize the ITI control is typical, we did not include it in the current study because we did not believe it was necessary. Briefly there are several reasons we felt this way. For starters, as the reviewers note, we chose our time period based on the identification of very specific neural correlates; the most direct way to disrupt these is to inactivate during these correlates. While it is straightforward to propose that inactivating in other time periods should not have any effect if our hypothesis is correct, it seems to us that this is not really correct. That is, inactivating in other periods – say right before or after the cues? – might also affect consolidation or other processes required for the learning. Yet this result would not invalidate the importance of our positive result from inhibition during the cue period. For this reason, we felt it was not critical to testing our hypothesis.

We also viewed the standard middle-of-the-ITI inhibition as a relatively trivial negative control in this particular situation, since the ITIs are several minutes long, and we already know from other studies that similar short periods of inactivation of OFC can be conducted without affecting learning and behavior. For example, in a Pavlovian over-expectation task we showed that inactivation during a period where OFC showed key neural correlates affected learning and behavior, whereas inactivation a couple of minutes later in the ITI did not.

Nevertheless, we realize that this concern will be shared by other readers, so we have added the below to the Discussion section to better explain the rationale behind this aspect of our experimental design:

“A second consideration to discuss is the timing of our manipulation and its specificity. We inhibited OFC only during cue presentation, precisely when we previously observed single unit correlates of cue-cue encoding; our design did not include control conditions in which similar inhibition was done before or after the cues or during the intertrial intervals. Thus, while we can conclude that OFC must be online at the time the neural correlates were observed, we cannot conclude that it does not need to be online at any other time during the first phase. Although OFC can be inactivated briefly between trials without affecting OFC-dependent learning and behavior (Takahashi et al., 2013), we would not be surprised if processing outside the strictly defined cue period – say immediately before or after – were also necessary for normal learning. Such a result would not invalidate what we have shown here, but it would show that learning is not fully completed during the external sensory events. While defining this critical period would be very interesting, it would require a succession of control groups that we did not deem practical (just after the cues, 20s after the cues, 40s after the cues, and so on).”

2) The discussion of attention in the learning of cue-cue value associations should be expanded as this is the key take away from the paper. Cues that the rats did not expect to be presented are likely learned about as they are attentionally salient. More of a discussion of the role of OFC in attentionally salient events and interaction with other areas feels more than a little appropriate here. For instance, OFC is known to be modulated during odd-ball tasks through input from the basal forebrain (Nguyen and Lin, 2014) and more recently work in primates has started to reveal that OFC encoding of value is attentionally gated (McGinty, et al., 2015). Similarly, how does this speak to learning models based on attention such as the Pearce-Hall model?

We agree this is a potentially important point, and we have expanded our Discussion section to try to address it more fully. As we note in the below, we recognize the evidence that the OFC is involved in and perhaps contributes to cue salience. However while we cannot rule disruption of this function out as the primary cause of our effect, it seems more parsimonious to us to propose that any such function is secondary to the associative learning role that OFC performs, as evidenced in the neural correlates that normally occur contemporaneous with our period of inhibition. This idea is consistent with ideas from Mackintosh as well as the Esber-Haselgrove successor to the Pearce-Hall and Mackintosh models, which proposes that cue salience varies with associative strength. Thus, our current results could be related to the role of OFC in regulating cue salience, but we would suggest it is because OFC is important for learning the sensory-sensory associations, not the other way around.

“A final consideration is that all rats received food cup approach shaping prior to preconditioning. It is possible that this changes the salience of the conditioning chamber context and causes increased attention to the cues during preconditioning, which may facilitate learning. If this is the case, then one interpretation of the current findings would be that OFC is necessary for normal acquisition of the associations during preconditioning because it contributes to this boost in attention, perhaps secondary to a role in processing the value of the context, rather than contributing to the encoding itself. This idea would be consistent with results showing correlates of risk (O'Neill and Schultz, 2010; Ogawa et al., 2013), uncertainty (van Duuren et al., 2009) and salience (Ogawa et al., 2013) in OFC neurons, as well as the finding that OFC value coding is attentionally gated (McGinty et al., 2016) and active in auditory oddball task (Nguyen and Lin, 2014). However while it is possible that inactivating OFC affected learning of the sensory-sensory associations by interfering with a primary role for OFC in supporting the context or even cue salience, it seems more parsimonious to us to propose that the role OFC plays in supporting salience is secondary to the associative learning role that OFC clearly supports. This idea is consistent with attentional models, both old and new, that link cue salience to associative strength – changes in cue associative strength drive changes in salience (Mackintosh, 1975; Pearce and Hall, 1980; Haselgrove et al., 2010; Esber and Haselgrove, 2011).”

3) The paper is brief and relies on a small amount of data. From what I can tell, the key analysis is Figure 2C, which is described in subsection “Orbitofrontal cortex activity is necessary for forming sensory associations”. The thing I don't totally get is that the finding relies on a positive effect in the control condition and a null effect in the experimental condition (subsection “Orbitofrontal cortex activity is necessary for forming sensory associations”). Strictly speaking, this is invalid – the difference between significant and non-significant is not in itself significant (it can be and often is, but isn’t necessarily so, Nieuwenhuis, Forstmann, and Wagenmakers, 2011). This is especially important to address because the control effect is marginally significant (p=0.046). Anyway, it looks like it probably is significant, but that ought to be reported. (Another reviewer during consultation surmised that you did do this analysis but they weren't quite sure "It looks like they have computed the interaction term and reported it 'however, there was a significant cue x group interaction (F(1,41)=4.57, p=0.04).'".

We regret that we did not make it clearer that we do report the interaction, as the reviewers surmised. The text has been updated to reflect this:

“There was a significant cue x group interaction (F_(1,41)_=4.57, *p*=0.04); ergo, the effect of cue during the probe test (A-C) did depend on group (YFP control – NpHR OFC inhibition)”.

4) The second issue is that a disruption of effect, while it does demonstrate causality, has multiple possible interpretations. It could be that the optogenetic effect is, like, super-distracting, or causes a headache or something, and that causes the animal to get distracted and not do very well on the task. There are probably ways to deal with this concern, like showing other things that aren't disrupted. But in this case, I want to see that because we don't want to assume causality from any abolition of effect.

We appreciate this issue but feel it is mitigated by several findings: (1) Identical stimulation in both lateral and medial OFC had no effect on economic choice behavior in our lab. These citations have been added. (2) If the stimulation was painful/aversive, then it would be expected to impair reward learning about these cues, since light occurred during each cue presentation. We observed intact reward conditioning in phase 2, and there was no difference between groups. We also did not see any effect of group during preconditioning when light was delivered. (3) Inactivation of lateral OFC is not sufficient to produce state-dependent learning. These have been added to the Discussion section:

“A third consideration is whether the effect of optogenetic inhibition could reflect something other than the inhibition of the neural correlates demonstrated in our earlier study. While we know from our control group, that simply delivering light during the cue period is not sufficient, perhaps inhibiting OFC neurons during this period distracts the rats, because it is rewarding or punishing or for some other reason. If this were the case, then we would have expected to see an impact of this on learning and responding to these cues in the later periods of the task. We did not see any evidence of this. Further we know from a number of prior studies that brief periods of OFC inhibition, similar to that employed here, often have minimal or no effects on ongoing behavior (Takahashi et al., 2013; Gardner et al., 2017; Gardner et al., 2018). These results suggest this is not, by itself, rewarding, painful or otherwise distracting. Finally, it has been shown that lateral OFC inactivation is not sufficient to produce state-dependent learning, which provides very strong evidence that it does not create a tangible experience in and of itself (Panayi and Killcross, 2014).”